# Numerical Analysis of Unsteady Hybrid Nanofluid Flow Comprising CNTs-Ferrousoxide/Water with Variable Magnetic Field

**DOI:** 10.3390/nano12020180

**Published:** 2022-01-06

**Authors:** Muhammad Sohail Khan, Sun Mei, Unai Fernandez-Gamiz, Samad Noeiaghdam, Said Anwar Shah, Aamir Khan

**Affiliations:** 1School of Mathematical Sciences, Jiangsu University, Zhenjiang 212013, China; sohailkhan8688@gmail.com (M.S.K.); shabnam8688@gmail.com (S.); 2Nuclear Engineering and Fluid Mechanics Department, University of the Basque Country UPV/EHU, Nieves Cano 12, 01006 Vitoria-Gasteiz, Spain; unai.fernandez@ehu.eus; 3Industrial Mathematics Laboratory, Baikal School of BRICS, Irkutsk National Research Technical University, 664074 Irkutsk, Russia; noiagdams@susu.ru; 4Department of Applied Mathematics and Programming, South Ural State University, Lenin Prospect 76, 454080 Chelyabinsk, Russia; 5Department of Basic Sciences and Islamiat, University of Engineering and Technology Peshawar, Peshawar 25120, Pakistan; anwarshah@uetpeshawar.edu.pk; 6Department of Pure and Applied Mathematics, University of Haripur, Peshawar 22620, Pakistan; aamir.khan@uoh.edu.pk

**Keywords:** ariable magnetic field, magnetic Reynold parameter, hybrid nano-fluid, PCM, BVP4C

## Abstract

The introduction of hybrid nanofluids is an important concept in various engineering and industrial applications. It is used prominently in various engineering applications, such as wider absorption range, low-pressure drop, generator cooling, nuclear system cooling, good thermal conductivity, heat exchangers, etc. In this article, the impact of variable magnetic field on the flow field of hybrid nano-fluid for the improvement of heat and mass transmission is investigated. The main objective of this study is to see the impact of hybrid nano-fluid (ferrous oxide water and carbon nanotubes) CNTs-$Fe_{3}O_{4}$, $H_{2}O$ between two parallel plates with variable magnetic field. The governing momentum equation, energy equation, and the magnetic field equation have been reduced into a system of highly nonlinear ODEs by using similarity transformations. The parametric continuation method (PCM) has been utilized for the solution of the derived system of equations. For the validity of the model by PCM, the proposed model has also been solved via the shooting method. The numerical outcomes of the important flow properties such as velocity profile, temperature profile and variable magnetic field for the hybrid nanofluid are displayed quantitatively through various graphs and tables. It has been noticed that the increase in the volume friction of the nano-material significantly fluctuates the velocity profile near the channel wall due to an increase in the fluid density. In addition, single-wall nanotubes have a greater effect on temperature than multi-wall carbon nanotubes. Statistical analysis shows that the thermal flow rate of ($Fe_{3}O_{4}$-SWCNTs-water) and ($Fe_{3}O_{4}$-MWCNTs-water) rises from 1.6336 percent to 6.9519 percent, and 1.7614 percent to 7.4413 percent, respectively when the volume fraction of nanomaterial increases from 0.01 to 0.04. Furthermore, the body force accelerates near the wall of boundary layer because Lorentz force is small near the squeezing plate, as the current being almost parallel to the magnetic field.

## 1. Introduction

Due to the wide range of engineering and scientific applications, the investigation of heat transfer and fluid movement between surfaces has been one of the most prominent research fields in recent times. Compared to nanofluids, hybrid nanofluids are expected to replace simple nanofluids for a variety of reasons, including wide absorption range, low volatility, low pressure, good thermal conductivity, low-pressure reduction, and low friction losses, solar energy, air conditioning applications, heat pipes, electronic cooling, biomedical engineering, ship, space, automotive industry, transformer cooling, and defence application are the few uses of hybrid nanofluid. In this regard, extensive research has been done to evaluate the heat and flow transfer properties between two plates. Mustafa et al. [1] studied the properties of mass, fluid movement, and heat transfer in two parallel plates. Their results reveal that increasing Schmidt number values reduce the concentration profile and increase the magnitude of the local Sherwood number. In addition, their results signify that the enhancement in the Prandtl number augments the Nusselt number. Dogonchi et al. [2] analyzed the transfer of flow and heat of the MHD graphene oxide/water nanoliquids in the presence of thermal radiation between two parallel plates and an established association between volume fraction, Nusselt number and temperature distribution. In addition, they discovered that the enhancement in the Reynolds number and extension ratio leads to an increase in the skin friction coefficient. Alizadeh et al. [3] investigated the properties of micropolar MHD fluid motion inside a channel loaded by nanoliquids bounded to the thermal radiation. Their results reveal that Nusselt number is a growing function which depends on the volume fraction of nanofluids and radiation parameters. Dib et al. [4] examined the flow and heat transfer of time-dependent nanoliquids between plates and concluded that differents kinds of nanoliquids play an important role in the fortification of heat transmission. Dogonchi et al. [5] studied the heat transfer of transient MHD flow of nanofluid at the surfaces influenced by thermal radiation, and concluded that a growing number of the radiation parameters leads to an increase in the temperature distribution and Nusselt number. Furthermore, they established a relationship between skin friction coefficient and Nusselt number, which increases with the increase in the magnetic parameter and volume fraction of the nanofluid. Sheikholeslami et al. [6] investigated the properties of heat transfer and the flow of nanofluids between two surfaces and in a revolving system. Their results reveal that in both suction and injection cases, the surface heat transfer rate enhances the volume fraction of nanomaterials, Reynolds number, and suction/injection parameter, and this reduces the power of the circulation parameter. In addition, their outcomes reveal that the Nusselt number has a direct relationship with the volume fraction of nanomaterial, suction/injection parameter, and Reynold number, while it possesses an inverse relationship with the power of rotation parameter. Mehmood and Ali [7] analyzed the heat transfer of the viscous MHD fluid flow fluid between two parallel surfaces. They took the influence of viscous dissipation in their analysis and discovered that the injection on the upper plate causes the temperature to rise while the magnetic field decreases the temperature. Furthermore, they reviewed that the Prandtl number with viscous dissipation enhance the temperature profile, while without viscous dissipation, the temperature decreases as the Prandtl number increases.

Fluids play a fundamental role in increasing the rate of heat transfer in many engineering applications, such as heat exchangers, fuel cells, etc. As we know, the regular fluids have a very low thermal conductivity in heat transferring. So, we need unusual, high thermal conductivity fluids to overcome this problem. This special type of fluid is called nanofluids. The practical application of nanofluid was first introduced by Choi [7]. The basic feature of nanofluids is that they have more thermal conductivity in comparison with regular fluids because the particles of metal nanometer-size are put into the liquid, which plays a valuable role in increasing the thermal conductivity. Most of the researchers have focused their analysis on fluid movement and heat transmission using normal fluid or nanofluid. Dogonchi and Ganji [8] studied the behaviour of buoyancy flow and heat transfer of MHD nanofluid on a stretching surface in the light of Brownian motion with thermal radiation. Their research explains that the temperature profile and the velocity of the fluid decrease when the radiation parameter is increased. Furthermore, they explained that the coefficient of skin friction increases with the increase of the magnetic parameter and decreases with the increase in the volume fraction of the nanofluid. Bhatti et al. [9] studied through a permeable extending wall the effect of entropy generation on non-Newtonian Eyring–Powell nanofluid, and discovered that the greater the effect of the suction parameter, the greater the velocity profile. In addition, their results show that thermophoresis parameters and Brownian motion significantly increase the temperature profile.

Chamkha et al. [10] analyzed the transient conjugate free convection, which applied to a half-circular pot with limited thickness solid walls containing Al${}_{2}$O${}_{3}$-Cu-water hybrid nanoliquids. They concluded that only a 5 percent increase in Al${}_{2}$O${}_{3}$-Cu nanomaterials indicates an increase in the average Nusselt number ranging from 4.9 to 5.4, while a 5 percent increase in Al${}_{2}$O${}_{3}$ nanomaterials increases the average Nusselt number from 4.9 to 5.36. In a square porous wall, MHD heat transfer and free convection flow are cooled and heated by sink or heat source, respectively, and loaded with a Cu-Al${}_{2}$O${}_{3}$-water hybrid nanofluid were investigated by Gorla et al. [11]. The results of this investigation signify that the average Nusselt number decreases significantly for hybrid suspension in case of change in the position of heat source. Furthermore, their outcomes reveal that the mean Nusselt number of hybrid suspensions is lower than that of Cu and Al${}_{2}$O${}_{3}$. The moving boundaries which produced squeezing flow play a significant role in polymer processing, hydrodynamical machines, and lubrication tools, etc. Jackson [12] studied the relationships between the squeezing fluid flow and loaded earing’s performance under the adhesive phenomena. Hayat et al. [13] examined the impact of couple stress fluid flow in the presence of the time-dependent magnetic field. Hayat et al. both [14,15] used the Buongiorno model of nanofluids, which indirectly tested certain nanoparticles. In the meantime, Salehi et al. [16] studied the squeezing hybrid nanofluid which can be formulated by putting the nanomaterial Fe${}_{3}$O${}_{4}$-MoS${}_{2}$ in the base fluid water and ethylene glycol. Acharya [17] analyzed the influence of radiation due to solar energy over Cu-Al${}_{2}$O${}_{3}$/water hybrid nanofluid within a channel. Ikram et al. [18] explored another fascinating feature of the hybrid nanofluid movement within a channel. Tayebi and Chamkha [19] studied free convection through the annulus between two elliptical cylinders filled with Cu-Al${}_{2}$O${}_{3}$/water hybrid nanofluid. It has been shown that heat transfer rate is more efficient if one uses Cu-Al${}_{2}$O${}_{3}$/water hybrid as compared to Al${}_{2}$O${}_{3}$/water nanofluid. Tayebi and Chamkha. [20] investigated the free convection of hybrid nanofluid in the eccentric annulus of horizontally cylindrical shape. Bilal and Taseer et al. [14,15] studied the (MHD) hybrid nanofluids (CNT-$Fe_{3}O_{4}$/$H_{2}O$) motion between horizontal channels through dilating and squeezing walls with thermal radiation. The flow is transient and laminar. The flow is not symmetric and the lower and upper walls vary in temperature and porosity. The suspension of single and multi-walls carbons nanotubes and Fe${}_{3}$O${}_{4}$ in nanofluids is exploited.

The purpose of the analysis was to examine the general quality of local thermal non-equilibrium (LTNE) and focus on the impact of nanomaterials (AA7075 and Cu) in the conventional fluids (methanol and NaAlg) using the Tiwari–Das model. They have investigated the heat transfer properties of nanofluids numerically from an engineering point of view at a stretching plate in the porous medium in [21,22]. Song et al. [23] analyzed the effect of Marangoni convection, thermal radiation, Soret and Dufour effects, viscous dissipation, nonlinear heat sink/source, and activation energy on MHD nanofluids motion produced by revolving the disk. Furthermore, they studied the effect of activation energy on the Darcy–Forchemer movement of Casson fluids, which include the suspension of titanium dioxide and graphene oxide nanomaterials containing 50 percent ethylene glycol as the conventional fluid in a porous medium [23,24,25,26,27]. The Marangoni convection of hybrid nanoliquid has been analyzed by Khan et al. [28,29], which is a combination of (MnZi, Fe${}_{2}$O, 4-NiZn, Fe${}_{2}$O${}_{4}$) nanomaterials and one conventional fluid (H${}_{2}$O), and the momentum equation updated through the inclusion of Darcy–Forchheimer in the porous medium. Due to heat transfer irreversibility, viscous dissipation irreversibility and mass transfer irreversibility, the entropy generation of the fluid flow is computed and analyzed through pertinent parameters.

A model of PVT unit containing PCM was analyzed by Khodadadi et al. [30]. The impact of different kinds of nanoliquids and NEPCMs at various concentrations on the system efficiency is estimated. The ZnO, SiC, Al${}_{2}$O${}_{3}$, MCNT (multi-walled carbon nanotube), Cu, and Ag nanomaterials are used inside the water with phase change particle concentrations of 0, 0.02, 0.04, by Sheikholeslami et al. [31,32]. The solar collector was investigated regarding the collector performance and irreversibility in the presence of the variable solar radiation. Along with these two main factors, performance factors are important functions that must be considered in order to reach the optimal design. Irreversibility is due to a decrease in temperature with an increase in wind speed due to a slight decrease in temperature of different zones. Uddin et al. [33] presented a novel model of bio-nano-transport, the impacts of first and second-order velocity slip, mass slip, heat slip, and gyro-tactic (torque-responsive) microorganism slip of bio-convective nanoliquid motion in the flowing plate with blowing tendency are numerically analyzed. Zohra et al. [34] observed the microfluidic devices based on the microfluidic associated technologies and microelectromechanical processes that have received a warm welcome in the field of science and engineering. This is a mathematical formulation which analyzes the fluid flow of steady forced convective revolving disk put into water-based nanoliquid with microorganisms. Thiyagarajan et al. [35] analyzed pleural outflow as an obstruction of the pleural cavity in the lung wall. The reversal of the lung and chest wall process causes pleural fluid to accumulate in the pleural space. Parietal lymphatic dilation is caused by an increase in pleural fluid. This proposed model has been designed to acquire new outcomes of respiratory tract infections which investigates the response with respect to the injection of transient mixed convection motion of visceral pleural liquid transports between two vertical porous sheets. M. K. Alam et al. [36] studied the behavior of mass and heat transmission on the time-dependent viscous squeezing flow along with changeable magnetic field. They used a revolving channel in their analysis. The proposed model has been updated through the inclusion of energy and variable magnetic field equations. Bilal et al. [37] and Khoshrouye [38] researched that the development of technologies in power engineering and microelectronics requires the improvement of efficient cooling systems. This advancement comprises the use of fins of notably changeable geometry within cavities to increment heat elimination from the heat producing process. It is considered that the fins are playing a significant role in augmenting heat transfer, so the proposed model has studied the impacts of several parameters on the transfer of the heat of embedded fins in cavities. In addition, the impacts range for some parameters on energy transmission.

Motivated by the prescribed literature review, it has been noticed that the analysis of hybrid nanofluids with $Fe_{3}O_{4}$, and SWCNT, MWCNTs, has so far not considered between two parallel porous plates with changeable magnetic fields. In addition, the impacts of changeable magnetic field on mass and heat transfer in rectangular coordinate system is a novel approach in the field. The introduction of hybrid nanofluids flow is very important due to its many applications in industrial and engineering processes. In this article, we are going to investigate the impacts of variable magnetic fields on the flow of hybrid nanofluids for the improvement of heat and mass transmission. In the composition of hybrid nanofluids, the ($Fe_{3}O_{4}$, single-wall carbon nanotubes and multi-wall carbon nanotubes) nanomaterials are used. Thermal radiation is also considered for high-temperature phenomena. The governing equations of the hybrid nanofluid are formulated under certain hypotheses, and solved numerically by (parametric continuation method) in MATLAB. The numerical outcomes of several emerging parameters skin frictions, Nusselt number, etc., are discussed through tables and graphs. In addition, it has been noticed that the thermal flow rate of ($Fe_{3}O_{4}$-SWCNTs-water) nanofluids rises from 1.6336 percent to 6.9519 percent when the volume fraction of nanomaterial increases from 0.01 to 0.04. In the same way, the thermal flow rate of ($Fe_{3}O_{4}$-MWCNTs-water) nanofluids rises from 1.7614 percent to 7.4413 percent when the volume fraction of nano-material increases from 0.01 to 0.04.

## 2. Formulation

We consider the flow of hybrid nanofluids between two horizontal infinite parallel plates, as depicted in Figure 1. The distance between two plates has taken as h(t) = $l\sqrt{1-at}$; furthemore, the upper plate move towards the lower plate with velocity $v=\frac{dh}{dt}$. We assumed that the temperature $T_{H}$ is constant at the upper plate, and the flow of hybrid nanofluid (CNTs-$Fe_{3}O_{4}$/$H_{2}O$) in the channel is incompressible and viscid. The physical characteristics of hybrid nanofluid flow CNTs-$Fe_{3}O_{4}$/$H_{2}O$) rely on time. As a result of the greater influence of variable magnetic fields, this type of behaviour of hybrid nanofluid is thought to be due to their magnetic characteristics. Single and multi-wall carbon nanotubes, on the other hand, have larger thermal conductivity. Subsequently, the hybrid nanofluid (CNTs-$Fe_{3}O_{4}$/$H_{2}O$) is developed by suspending a new volume fraction of CNT ($\Phi_{2}$ = 0.5) into the originally formulated ferrofluid ($Fe_{3}O_{4}$/$H_{2}O$). The mathematical formulation of the aforementioned hybrid nanofluids by continuity, momentum, magnetic field and energy conservation equations are as follows:

Continuity equation [15]:(1)$$\frac{\partial u}{\partial x}+\frac{\partial v}{\partial x}=0,$$

Momentum equations [15,21]:(2)$$\begin{array}{c} \frac{\partial u}{\partial t}+u\frac{\partial u}{\partial x}+v\frac{\partial u}{\partial y}=-\frac{1}{\rho_{hnf}}\frac{\partial P}{\partial x}+\frac{\mu_{hnf}}{\rho_{hnf}}\left(\frac{\partial^{2}u}{\partial x^{2}}+\frac{\partial^{2}u}{\partial y^{2}}\right)-\frac{b_{2}\sigma_{hnf}}{\rho_{hnf}}\left(\frac{\partial b_{1}}{\partial y}-\frac{\partial b_{2}}{\partial x}\right) \end{array}$$
(3)$$\begin{array}{c} \frac{\partial v}{\partial t}+u\frac{\partial v}{\partial x}+v\frac{\partial v}{\partial y}=-\frac{1}{\rho_{hnf}}\frac{\partial P}{\partial y}+\frac{\mu_{hnf}}{\rho_{hnf}}\left(\frac{\partial^{2}v}{\partial x^{2}}+\frac{\partial^{2}v}{\partial y^{2}}\right)-\frac{b_{1}\sigma_{hnf}}{\rho_{hnf}}\left(\frac{\partial b_{1}}{\partial y}-\frac{\partial b_{2}}{\partial x}\right) \end{array}$$

Maxwell Equations [21,22]:(4)$$\frac{\partial b_{1}}{\partial t}=u\frac{\partial b_{2}}{\partial y}+b_{2}\frac{\partial u}{\partial y}-v\frac{\partial b_{1}}{\partial y}-b_{1}\frac{\partial v}{\partial y}+\frac{1}{\sigma_{hnf}\mu_{e}}\left(\frac{\partial^{2}b_{1}}{\partial x^{2}}+\frac{\partial^{2}b_{1}}{\partial y^{2}}\right)$$
(5)$$\frac{\partial b_{2}}{\partial t}=v\frac{\partial b_{1}}{\partial x}+b_{1}\frac{\partial v}{\partial x}-u\frac{\partial b_{2}}{\partial x}-b_{2}\frac{\partial u}{\partial x}+\frac{1}{\sigma_{hnf}\mu_{e}}\left(\frac{\partial^{2}b_{2}}{\partial x^{2}}+\frac{\partial^{2}b_{2}}{\partial y^{2}}\right)$$

The Energy Equation [15]:(6)$$\begin{array}{cc} \frac{\partial T}{\partial t}+u\frac{\partial T}{\partial x}+v\frac{\partial T}{\partial y} & =\frac{\kappa_{hnf}}{{\left(\rho C_{p}\right)}_{hnf}}\left(\frac{\partial^{2}T}{\partial x^{2}}+\frac{\partial^{2}T}{\partial y^{2}}\right)\\ & +\frac{\mu_{hnf}}{{\left(\rho C_{p}\right)}_{hnf}}\left(4{\left(\frac{\partial v}{\partial y}\right)}^{2}+{\left(\frac{\partial u}{\partial y}\right)}^{2}+{\left(\frac{\partial v}{\partial x}\right)}^{2}+2\frac{\partial u}{\partial y}\frac{\partial v}{\partial x}\right) \end{array}$$
where $b_{1},b_{2}$ are the components of magnetic field, ${\left(\rho C_{p}\right)}_{hnf}$ is the heat capacity of the hybrid nanofluid, *P* is fluid pressure, *T* is the temperature, $\rho_{hnf}$ is fluid density of hybrid nanofluid, $\sigma_{hnf}$ is electrical conductivity of hybrid nanofluid, $\mu_{hnf}$ is kinematic viscosity of hybrid nanofluid.

Nanofluid are defined as [15]:(7)$$\begin{array}{cc} \; & \nu_{hnf}=\frac{\mu_{h}nf}{\rho_{h}nf},\;\frac{\rho_{hnf}}{\rho_{f}}=\left(1-\phi_{2}\right)\left(\left(1-\Phi_{1}+\frac{\Phi_{1}\rho_{MS}}{\rho_{f}}\right)+\frac{\Phi_{2}\rho_{CNT}}{\rho_{f}}\right),\\ & \frac{\kappa_{hnf}}{\kappa_{bf}}=\frac{\kappa_{CNT}+\left(m-1\right)\kappa_{bf}+\phi_{2}\left(\kappa_{bf}-\kappa_{CNT}\right)}{\kappa_{CNT}+\left(m-1\right)\kappa_{bf}+\phi_{2}\left(m-1\right)\left(\kappa_{bf}-\kappa_{CNT}\right)}\\ & \frac{\kappa_{bf}}{\kappa_{f}}=\left(\frac{\kappa_{MS}+\left(m-1\right)\kappa_{f}+\phi_{1}\left(\kappa_{f}-\kappa_{MS}\right)}{\kappa_{MS}+\left(m-1\right)\kappa_{f}+\phi_{1}\left(m-1\right)\left(\kappa_{f}-\kappa_{MS}\right)}\right)\;\\ & \;\frac{{\left(\rho Cp\right)}_{hnf}}{{\left(\rho C_{p}\right)}_{f}}=\left(1-\phi_{2}\right)\left(\left(1-\Phi_{1}+\frac{\Phi_{1}{\left(\rho C_{p}\right)}_{MS}}{{\left(\rho C_{p}\right)}_{f}}\right)+\frac{\Phi_{2}{\left(\rho C_{p}\right)}_{CNT}}{{\left(\rho C_{p}\right)}_{f}}\right),\\ & \frac{\sigma_{hnf}}{\sigma_{f}}=1+\frac{3\left(\frac{\sigma_{p}}{\sigma_{f}}-1\right)\phi_{1}}{\left(\frac{\sigma_{p}}{\sigma_{f}}+2\right)-\left(\frac{\sigma_{p}}{\sigma_{f}}-1\right)\phi },\;\frac{\mu_{hnf}}{\mu_{f}}=\frac{1}{{\left(1-\phi_{1}\right)}^{2.5}{\left(1-\phi_{2}\right)}^{2.5}}, \end{array}$$
with $\kappa_{hnf}$ is the thermal conductivity of hybrid nanofluid, $\kappa_{bf}$ is the thermal conductivity of the $Fe_{3}O_{4}$-nanofluid, and $\phi_{1}$, $\Phi_{2}$ are the volume fraction of CNTs.

## 3. Boundary Conditions

The boundary conditions of the proposed model are as follows [15]:(8)$$\begin{array}{cc} \; & u=0,\;v=-\frac{dh}{dt},\;b_{1}=\frac{axM_{0}}{2(1-at)},\;b_{2}=\frac{axM_{0}}{2\sqrt{1-at}},\;T=T_{H}\;\;at\;y=h\left(t\right)\\ & u=0,\;v=0,\;T=0,\;b_{1}=b_{2}=0,\;\;\;at\;y=0 \end{array}$$

The following similarity transformations have been used for reducing a system of PDEs (1–6) into a non-linear system of ODEs [21],
(9)$$\begin{array}{cc} \; & u=\frac{ax}{2(1-at)}f^{\prime }\left(\eta \right),\;v=-\frac{al}{2\sqrt{1-at}}f\left(\eta \right),\;b_{1}=\frac{axM_{0}}{2(1-at)}K^{\prime }\left(\eta \right),\;\\ & b_{2}=-\frac{alM_{0}}{2\sqrt{1-at}}K\left(\eta \right),\;\eta =\frac{y}{l\sqrt{1-at}},\;\theta \left(\eta \right)=\frac{T}{T_{H}},\; \end{array}$$

Therefore, Equation (Equation 1) of the model has satisfied automatically, and the reduced forms of the remaining Equations (2)–(6) are as follows:(10)$$\begin{array}{cc} f^{\prime \prime \prime \prime } & =\frac{\rho_{hnf}}{\rho_{f}}\frac{\mu_{f}}{\mu_{hnf}}\left(S\left(3f^{\prime \prime }+\eta f^{\prime \prime }+f^{\prime }f^{\prime \prime }-ff^{\prime \prime \prime }\right)\right)+Ha^{2}SRe_{m}{\left(\frac{\sigma_{hnf}}{\sigma_{f}}\right)}^{2}\frac{\mu_{f}}{\mu_{hnf}}\left(2KK^{\prime }+fK^{2}\right)\\ & +\eta Ha^{2}S^{2}Re_{m}^{2}{\left(\frac{\sigma_{hnf}}{\sigma_{f}}\right)}^{3}\frac{\mu_{f}}{\mu_{hnf}}\left(K^{2}+\eta KK^{\prime }-fKK^{\prime }+f^{\prime }K^{2}\right)\\ & -Ha^{2}S^{2}Re_{m}^{2}{\left(\frac{\sigma_{hnf}}{\sigma_{f}}\right)}^{3}\frac{\mu_{f}}{\mu_{hnf}}\left(fK^{2}+\eta fKK^{\prime }-f^{2}KK^{\prime }+ff^{\prime }K^{2}\right), \end{array}$$
(11)$$K^{\prime \prime }=Re_{m}S\frac{\sigma_{hnf}}{\sigma_{f}}\left(K+\eta K^{\prime }-fK^{\prime }+f^{\prime }K\right),$$
(12)$$\begin{array}{cc} \theta^{\prime \prime } & =SPr\frac{{\left(\rho Cp\right)}_{hnf}}{{\left(\rho C_{p}\right)}_{f}}\frac{\kappa_{f}}{\kappa_{hnf}}\left(\eta \theta^{\prime }-\theta^{\prime }f\right)-\frac{\mu_{hnf}}{\mu_{f}}\frac{\kappa_{f}}{\kappa_{hnf}}PrEc\left(4\delta f^{\prime 2}+f^{\prime \prime 2}\right), \end{array}$$
and the boundary conditions in the reduced form as follows,
(13)$$\begin{array}{cc} \; & f\left(0\right)=0,\;f^{\prime }\left(0\right)=0,\;K\left(0\right)=0,\;\theta \left(0\right)=0,\\ & f\left(1\right)=1,\;f^{\prime }\left(1\right)=0,\;K\left(1\right)=1,\;\theta \left(1\right)=1, \end{array}$$
where $Ha^{2}=\frac{l^{2}M_{0}^{2}a\sigma_{f}}{\rho_{f}\nu_{f}}$ Hartmann number, $S=\frac{al^{2}}{2\nu_{f}}$ squeeze number, $Re_{m}=\sigma_{f}\nu_{f}\mu_{e}$ Rynold’s magnetic parameter, $Pr=\frac{\nu_{f}{\left(\rho Cp\right)}_{f}}{\kappa_{f}}$ Prandtl number, $Ec=\frac{a^{2}}{4{\left(Cp\right)}_{f}T_{H}{\left(1-at\right)}^{2}}$ Eckert number, $\delta =\frac{l^{2}\left(1-at\right)}{a^{2}}$.

Emerging physical parameters in the reduced form of system are the Nusselt number and skin friction coefficient, and can be defined as,
(14)$$C_{f}=\frac{\mu_{nf}}{\rho_{nf}\nu_{l}^{2}x}{\left(\frac{\partial u}{\partial y}\right)}_{y=h(t)},\;\;\;\;N_{u}=-\frac{a\kappa_{nf}{\left(\frac{\partial T}{\partial y}\right)}_{z=h(t)}}{k_{f}T_{H}},$$

In case of Equation (Equation 16), we get
(15)$$R_{1}^{2}C_{f}\frac{\rho_{hnf}}{\rho_{f}}\frac{\mu_{f}}{\mu_{hnf}}=f^{\prime \prime }\left(1\right),\;\;\;\;-\theta^{\prime }\left(1\right)=\frac{\kappa_{f}}{\kappa_{hnf}}R_{2}N_{u}.$$

## 4. Numerical Solution by PCM

In this section, optimal choices of continuation parameters are made through the algorithm of PCM [29] for the solution of non-linear Equations (10)–(12) with boundary conditions in Equation (Equation 13):**Step 1: First order of ODE**To transform the Equations (10)–(12) into the first order of ODEs, consider the following
(16)$$\begin{array}{cc} \; & f=t_{1},\;f^{\prime }=t_{2},\;f^{\prime \prime }=t_{3},\;f^{\prime \prime \prime }=t_{4}\\ & K=t_{5},\;K^{\prime }=t_{6},\;\theta =t_{7},\;\theta^{\prime }=t_{8} \end{array}$$
putting these transformations in Equations (10)–(12), which becomes
(17)$$\begin{array}{cc} t_{4}^{\prime }= & \frac{\rho_{hnf}}{\rho_{f}}\frac{\mu_{f}}{\mu_{hnf}}\left(S\left(3t_{3}+\eta t_{3}+t_{2}t_{3}-t_{1}t_{4}\right)\right)+Ha^{2}SRe_{m}{\left(\frac{\sigma_{hnf}}{\sigma_{f}}\right)}^{2}\frac{\mu_{f}}{\mu_{hnf}}\left(2t_{5}t_{6}+t_{1}t_{5}^{2}\right)\\ & +\eta Ha^{2}S^{2}Re_{m}^{2}{\left(\frac{\sigma_{hnf}}{\sigma_{f}}\right)}^{3}\frac{\mu_{f}}{\mu_{hnf}}\left(t_{5}^{2}+\eta t_{5}t_{6}-t_{1}t_{5}t_{6}+t_{2}t_{5}^{2}\right)\\ & -Ha^{2}S^{2}Re_{m}^{2}{\left(\frac{\sigma_{hnf}}{\sigma_{f}}\right)}^{3}\frac{\mu_{f}}{\mu_{hnf}}\left(t_{1}t_{5}^{2}+\eta t_{1}t_{5}t_{6}^{\prime }-t_{1}^{2}t_{5}t_{6}+t_{1}t_{2}t_{5}^{2}\right), \end{array}$$
(18)$$t_{6}^{\prime }=Re_{m}S\frac{\sigma_{hnf}}{\sigma_{f}}\left(t_{5}+\eta t_{6}-t_{1}t_{6}+t_{2}t_{5}\right),$$
(19)$$\begin{array}{c} t_{8}^{{}^{\prime }}=SPr\frac{{\left(\rho Cp\right)}_{hnf}}{{\left(\rho C_{p}\right)}_{f}}\frac{\kappa_{f}}{\kappa_{hnf}}\left(\eta t_{8}-t_{1}t_{8}\right)-\frac{\mu_{hnf}}{\mu_{f}}\frac{\kappa_{f}}{\kappa_{hnf}}PrEc\left(4\delta t_{2}^{2}+t_{3}^{2}\right), \end{array}$$
and the boundary conditions becomes
(20)$$\begin{array}{cc} \; & t_{1}\left(0\right)=0,\;t_{1}\left(1\right)=1,\;t_{2}\left(0\right)=0,\;t_{2}\left(1\right)=0,\\ & t_{5}\left(0\right)=0,\;t_{5}\left(1\right)=1,\;t_{7}\left(0\right)=0,\;t_{7}\left(1\right)=1, \end{array}$$**Step 2: Introducing of parameter p and we obtained ODEs in a p-parameter group**To get ODE’s in a p-parameter group, let we know p-parameter in Equations (17)–(19) and therefore,
(21)$$\begin{array}{cc} t_{4}^{\prime }= & \frac{\rho_{hnf}}{\rho_{f}}\frac{\mu_{f}}{\mu_{hnf}}\left(S\left(3t_{3}+\eta t_{3}+t_{2}t_{3}-t_{1}\left(t_{4}-1\right)q\right)\right)+Ha^{2}SRe_{m}{\left(\frac{\sigma_{hnf}}{\sigma_{f}}\right)}^{2}\frac{\mu_{f}}{\mu_{hnf}}\left(2t_{5}t_{6}+t_{1}t_{5}^{2}\right)\\ & +\eta Ha^{2}S^{2}Re_{m}^{2}{\left(\frac{\sigma_{hnf}}{\sigma_{f}}\right)}^{3}\frac{\mu_{f}}{\mu_{hnf}}\left(t_{5}^{2}+\eta t_{5}t_{6}-t_{1}t_{5}t_{6}+t_{2}t_{5}^{2}\right)\\ & -Ha^{2}S^{2}Re_{m}^{2}{\left(\frac{\sigma_{hnf}}{\sigma_{f}}\right)}^{3}\frac{\mu_{f}}{\mu_{hnf}}(t_{1}t_{5}^{2}+\eta t_{1}t_{5}t_{6}^{\prime }-t_{1}^{2}t_{5}t_{6}+t_{1}t_{2} \end{array}$$
(22)$$t_{6}^{\prime }=Re_{m}S\frac{\sigma_{hnf}}{\sigma_{f}}\left(t_{5}+\eta t_{6}-t_{1}\left(t_{6}-1\right)q+t_{2}t_{5}\right),$$
(23)$$\begin{array}{c} t_{8}^{{}^{\prime }}=SPr\frac{{\left(\rho Cp\right)}_{hnf}}{{\left(\rho C_{p}\right)}_{f}}\frac{\kappa_{f}}{\kappa_{hnf}}\left(\eta \left(t_{8}-1\right)q-t_{1}t_{8}\right)-\frac{\mu_{hnf}}{\mu_{f}}\frac{\kappa_{f}}{\kappa_{hnf}}PrEc\left(4\delta t_{2}^{2}+t_{3}^{2}\right), \end{array}$$**Step 3: Differentiation by p, reaches the following system w.r.t the sensitivities to the parameter-p**Differentiating the Equations (21)–(23) w.r.t by *p*
(24)$$d_{1}^{\prime }=h_{1}d_{1}+e_{1}$$
where $h_{1}$ is the coefficient matrix, $e_{1}$ is the remainder and $d_{1}=\frac{dp_{i}}{d\tau }$, 1≤i≤8.**Step 3: Cauchy Problem**(25)$$d_{1}=y_{1}+a1v_{1},$$
where $y_{1}$, $v_{1}$ are vector functions. By resolving the two Cauchy problems for every component. We are satisfied then automatically to ODE’s
(26)$$e_{1}+h_{1}\left(a1v_{1}+y_{1}\right)={\left(a1v_{1}+y_{1}\right)}^{{}^{\prime }}$$
and left the boundary conditions.**Step 4: Using by Numerical Solution**An absolute scheme has been used for the resolution of the problem
(27)$$\frac{v_{1}^{i+1}-v_{1}^{i}}{▵\eta }=h_{1}v_{1}^{i+1}$$
(28)$$\frac{y^{i+1}-y^{i}}{▵\eta }=h_{1}y^{i+1}+e_{1}$$**Step 5: Taking of the corresponding coefficients**As given boundaries are usually applied for $p_{i}$, where 1≤i≤8, for the solution of ODEs, we required to apply $d_{2}=0$, which seems to be in matrix form as
(29)$$l_{1}.d_{1}=0\;\mathrm{or}\;l_{1}.\left(a1v_{1}+y_{1}\right)=0$$
where $a1=\frac{-l_{1}.y_{1}}{l_{1}.v_{1}}$

## 5. Results and Discussions

This paper investigates the hybrid nano-fluid flow between two parallel and squeezing plates under the influence of a variable magnetic field. The heat and mass transfer phenomena are also considered. The governing system of the hybrid nanofluid flow eqautions is converted into a non-linear system of ODEs through similarity transformations. The impacts of various physical parameters, including Hartmann number (Ha), magnetic Reynolds number ($Re_{m}$), Prandtl number (Pr), squeezing parameter (S), Eckert number (Ec), and hybrid nano-particle volume fraction ($\phi_{1}$, $\phi_{2}$) have been investigated in the context of heat transfer and fluid flow properties. The statistics in Table 1 provide complete information about the thermophysical properties of nanomaterials. It is mandatory to mention that our proposed model produces a nice results when compared with the results of the models available in the current literature, and the comparison has been shown in Table 2. Table 3, Table 4, Table 5, Table 6 and Table 7 illustrates the numerical outcomes of two important flow parameters skin friction and Nusselt number, which are obtained through two different numerical schemes(BVP4C and PCM). Table 8 displays quantitatively the impact of various flow parameters for different types of hybrid nano-fluid.

In Figure 2a, it has been observed that the velocity profile for different values of the squeezing parameter *S* increases the velocity profile more near the channel due to the rising force in the horizontal direction. This increment in force has a resistance force near the centre, which decreases the horizontal velocity after the central region. Figure 2b depicts that the growing value of squeezing parameter *S* improves the velocity profile, it is because the velocity gradient is falling near the channel wall for η<0.5, while vertical velocity $f^{\prime }\left(\eta \right)$ is rising due to narrowing the channel of the flow for the value η>0.5. The velocity profile of the flow is plotted in Figure 3a,b, illustrates that Ha and $Re_{m}$ are rising. Furthermore, it explains that the increasing value of the Hartmann number raises the velocity profile for η<0.6, and reduces it for η>0.6. Cross-flow behaviour for the velocity profile has been observed at the centre of the channel wall. The effect of the nano-material volume friction is depicted in Figure 4. It is noticed that a rise in the quantity of nanomaterials significantly fluctuates the velocity profile near the channel wall due to the increase in the density of the nano-materials. Moreover, the velocity of SWCNTs is slightly higher than that of MWCNTs, due to the low-density values of SWCNTs.

Figure 5a displays the influence of the squeezing parameter *S* on the magnetic field profile G(η), which illustrates that the magnetic profile becomes parabolic for the various value of the squeezing parameter. Body force is stronger near the bottom of the corresponding main wall because the Lorentz force is smaller near the squeezing plate (because the current is approximately parallel to the magnetic field). Initially, for the x-component of velocity, the velocity decline is identified, but starts augmenting as η→1 maximum value of $f^{\prime }\left(\eta \right)$ is noticed in the middle. In the study of the magnetic profile, we have observed that the magnetic profile G(η) is falling in the vicinity of the upper plate when the squeezing parameter increases. Figure 5b shows the falling behaviour of the magnetic Reynold’s parameter $Re_{m}$, which defines the ratio of fluid flux to the magnetic diffusivity. This parameter therefore is an instrumental in determining the diffusion of magnetic field along streamlines. The variation in $Re_{m}$ has direct effect on G(η). As the increase in magnetic Reynold’s number increases, the opposing force of the axial velocity decreases the axial velocity of the flow. In our study, less importance has been placed on diffusion, and most of the analysis is based on the behaviour of the magnetic field during the flow.

Figure 6 illustrates the impact of $\Phi_{2}$ on the magnetic profile, which signifies that an opposing force is generated due to magnetic fields, as well as an improvement in the fluid viscosity due to the suspension of nanomaterials concentration. This is the basic reason for the fall in the magnetic field. The outcomes reflect that, as the concentration of nanomaterials rises, the magnetic field decreases. Figure 7a explains the impact of growing Eckert number on the temperature profile. The Eckert number is the ratio of kinetic energy to the boundary layer enthalpy difference and is used to explained heat dissipation. An abrupt surge for the temperature profile in the vicinity of the middle line has been noticed, as we know Ec simply the ratio of specific heat to thermal conductivity, so it raises the temperature profile considerably, and θ(η) profile becomes parabolic, which gives the maximum value at the middle of the channel. The heat transmission of (MWCNTs, $Fe_{3}O_{4}$ and $H_{2}O$) is slightly more than (SWCNTs, $Fe_{3}O_{4}$ and $H_{2}O$). The fluctuation that occurs in temperature profile due to the rising value of nanomaterials volume fraction $\Phi_{2}$ has been displayed in Figure 8. A rise in the volume fraction of nanomaterials augments heat transmission and generation of heat, which grow the thickness of the thermal layer. In the case of diverging channels, a significant increase in temperature is observed. Nearly the same values for both MWCNT and SWCNT with $H_{2}O$ and $Fe_{3}O_{4}$ have been observed. The velocity slip parameter significantly affects the temperature profile. Figure 9 and Figure 10 are drawn to investigate the effects of *S*, Ha, and *S*, Ec on skin friction and Nusselt number.

## 6. Concluding Remarks

The main objective of this article is to analyze the effects of a changeable magnetic field on the hybrid nanofluids flow between the squeezing parallel plates so as to improve the mass and heat transfer profile. Hybrid nanofluid (ferrous oxide water and carbon nanotubes) CNTs-$Fe_{3}O_{4}$, $H_{2}O$ have been chosen for this purpose. The governing equation of hybrid nanofluids flow including, momentum equaiton, energy equations, and magnetic field equations have been converted into systems of highly nonlinear ODEs thorugh similarity transformations, and subsequently solved by the parametric continuation method (PCM). For the validity of the numerical solution, the proposed model is also solved by the shooting method.

The main findings of this study are as follows:Statistical analysis shows that the thermal flow rate of ($Fe_{3}O_{4}$-SWCNTs-water) rises from 1.6336 percent to 6.9519 percent when the volume fraction of nano-material increases from 0.01 to 0.04 is shown in Table 4.It is obvious from Table 3 that the thermal flow rate of ($Fe_{3}O_{4}$-MWCNTs-water) rises from 1.7614 percent to 7.4413 percent when the volume fraction of nano-material increases from 0.01 to 0.04.Increase in the volume friction of the value of nano-material significantly fluctuates the velocity profile near the channel wall due to the increase in the fluid density.The increasing value of the squeezing parameter comparatively improves the velocity profile near the channel wall more than the center.By increasing the squeezing speed of the upper plate improve the velocity profile, it is because the velocity gradient is falling near the channel wall for η<0.5, while vertical velocity $f^{\prime }\left(\eta \right)$ is rising, due to narrowing the channel of the flow for the value η>0.5.Body force is stronger near the bottom of the corresponding main wall, because the Lorentz force is smaller near the squeezing plate (because the current is approximately parallel to the magnetic field). Initially, for the x-component of velocity, the velocity decline is identified, but starts augmenting as η→1 maximum value of $f^{\prime }\left(\eta \right)$ is noticed at the middle.It has been noticed that the magnetic Reynolds number increases opposing force of the axial velocity, which decreases the axial velocity of the flow.Strengthening the magnetic Reynold’s number increases the strength of the opposing force of the axial velocity, which decreases the axial velocity of the flow.

## 7. Future Work

This problem can be solved in the future in three-dimensional geometry. In addition, the effect of variable magnetic fields on the physical properties of heat and mass transfer and fluids can be interesting work to investigate.

## Figures and Tables

**Figure 1 nanomaterials-12-00180-f001:**
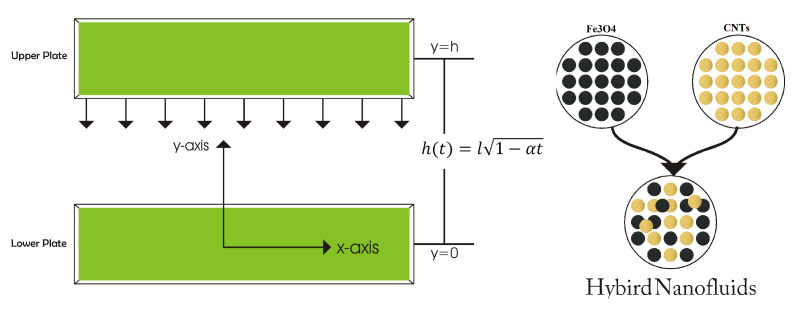
Geometry of the problem with coordinate system.

**Figure 2 nanomaterials-12-00180-f002:**
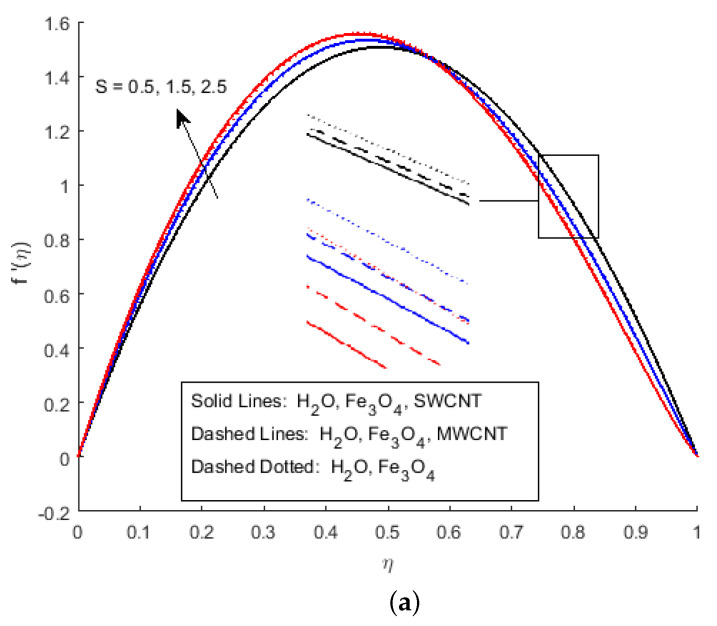
Effect of f(η) and $f^{\prime }\left(\eta \right)$ for (**a**) S>0, (**b**) S<0 and fixed values of $Re_{m}=1.5,$$Ha=0.4,Pr=6.2,Ec=0.5,\delta =0.4,\Phi_{1}=0.02,\Phi_{2}=0.5,m=0.3$.

**Figure 3 nanomaterials-12-00180-f003:**
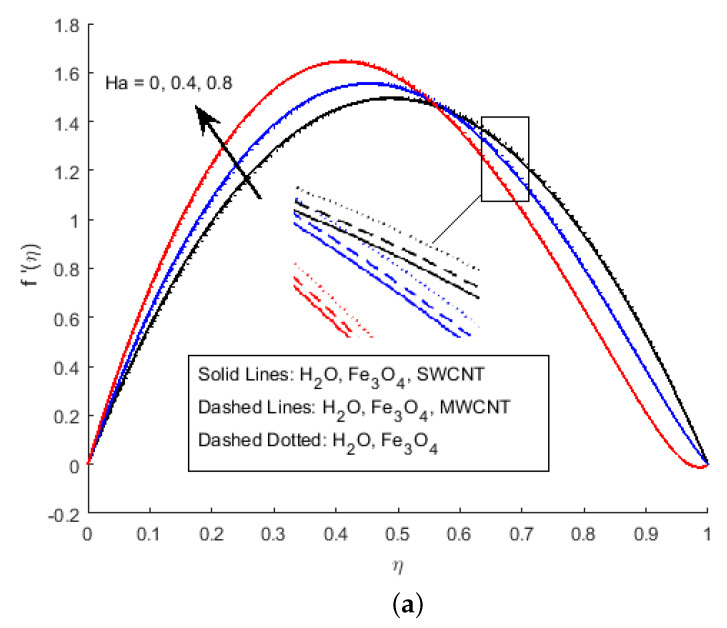
Effect of $f^{\prime }\left(\eta \right)$ for (**a**) Ha, (**b**) $Re_{m}$ and fixed values of $S=2.5,Pr=6.2,Ec=0.5,\delta =0.4,\Phi_{1}=0.02,\Phi_{2}=0.5,m=0.3$.

**Figure 4 nanomaterials-12-00180-f004:**
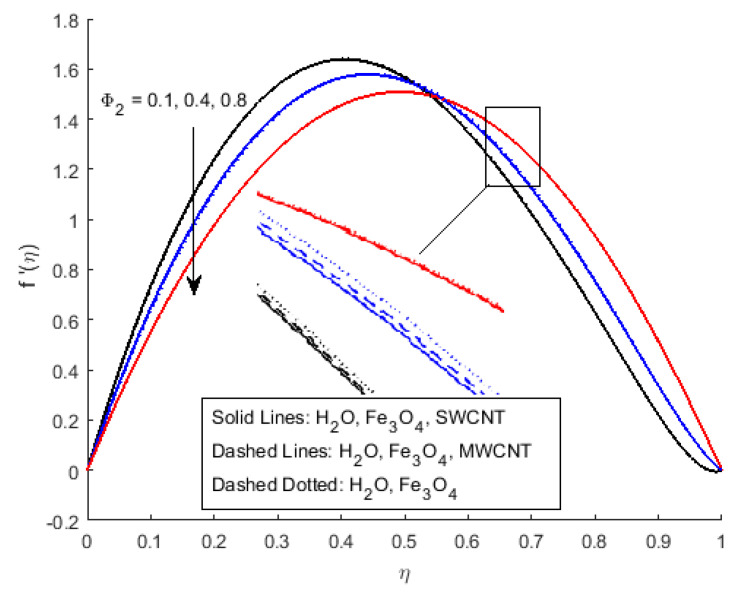
Effect of $f^{\prime }\left(\eta \right)$ for $\Phi_{2}$ and fixed values of $S=1.5,Re_{m}=2.5,Ha=0.4,Pr=6.2,Ec=0.5,\delta =0.4,\Phi_{1}=0.02,m=0.3$.

**Figure 5 nanomaterials-12-00180-f005:**
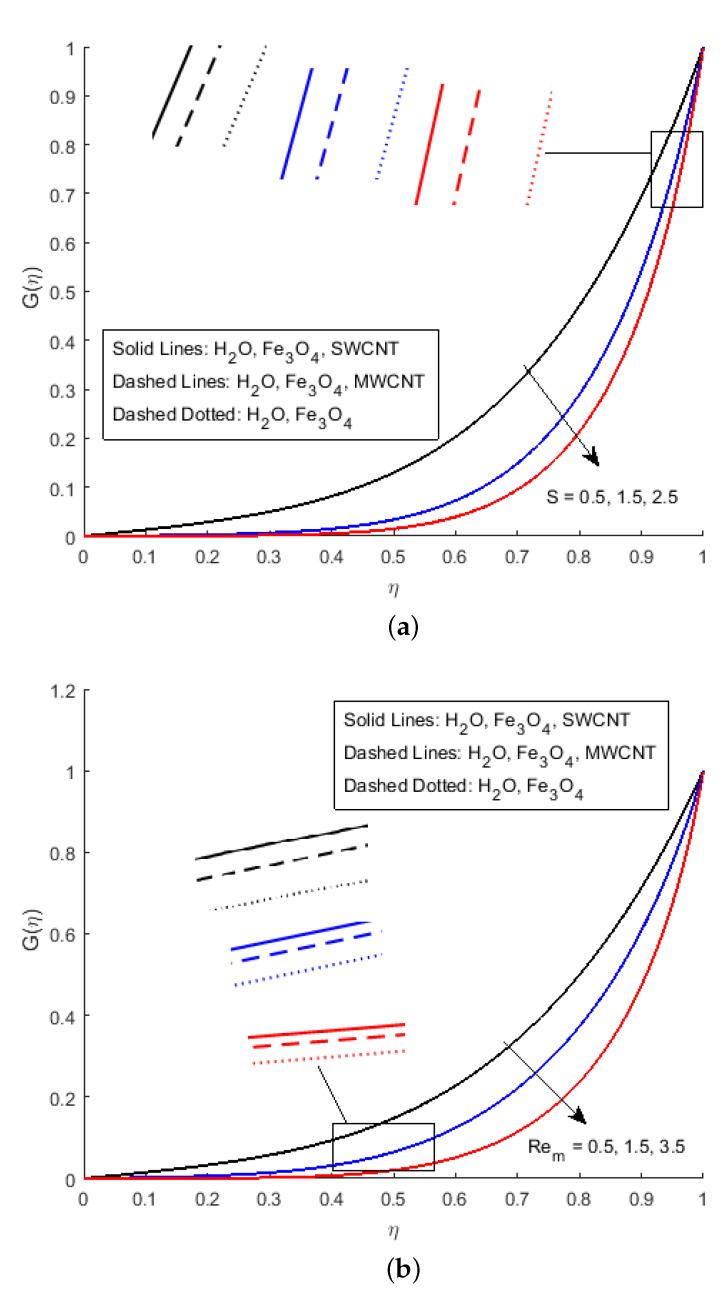
Effect of G(η) for (**a**) *S*, (**b**) $Re_{m}$ and fixed values of $Ha=1.4,Pr=6.2,Ec=0.5,\delta =0.4,\Phi_{1}=0.02,\Phi_{2}=0.5,m=0.3$.

**Figure 6 nanomaterials-12-00180-f006:**
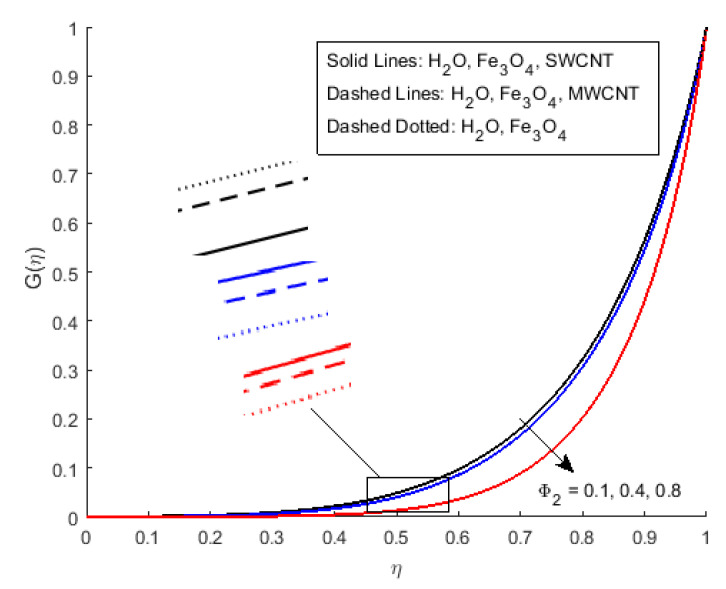
Effect of G(η) for $\Phi_{2}$ and fixed values of $S=1.5,Re_{m}=1.5,Ha=0.4,Pr=6.2,Ec=0.5,\delta =0.4,\Phi_{1}=0.02,m=0.3$.

**Figure 7 nanomaterials-12-00180-f007:**
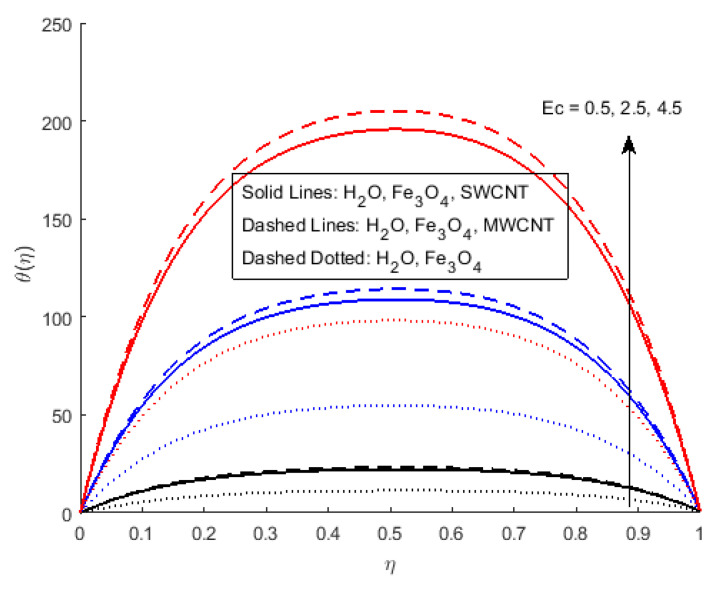
Effect of θ(η) for Ec and fixed values of $S=0.5,Re_{m}=1.5,Pr=6.2,Ha=0.4,\delta =0.4,\Phi_{1}=0.02,\Phi_{2}=0.5,m=0.3$.

**Figure 8 nanomaterials-12-00180-f008:**
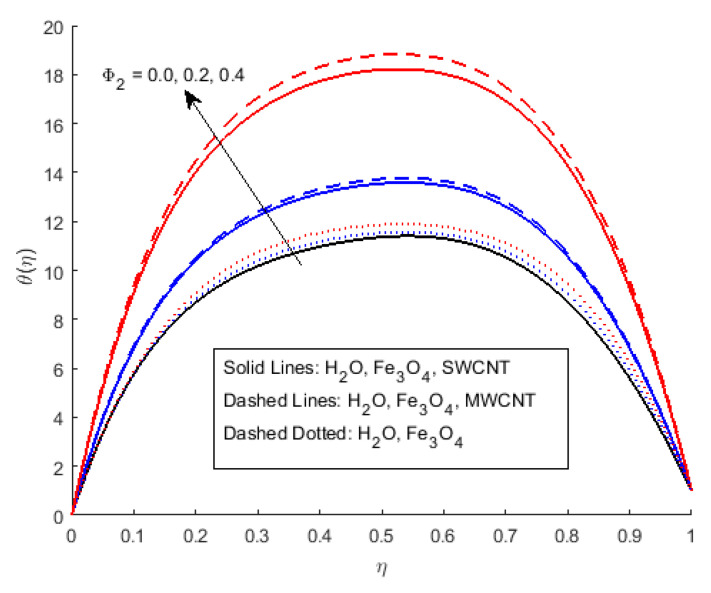
Effect of θ(η) for $\Phi_{2}$ and fixed values of $S=0.5,Re_{m}=2.5,Ha=0.4,Pr=6.2,Ec=0.5,\delta =0.4,\Phi_{1}=0.02,m=0.3$.

**Figure 9 nanomaterials-12-00180-f009:**
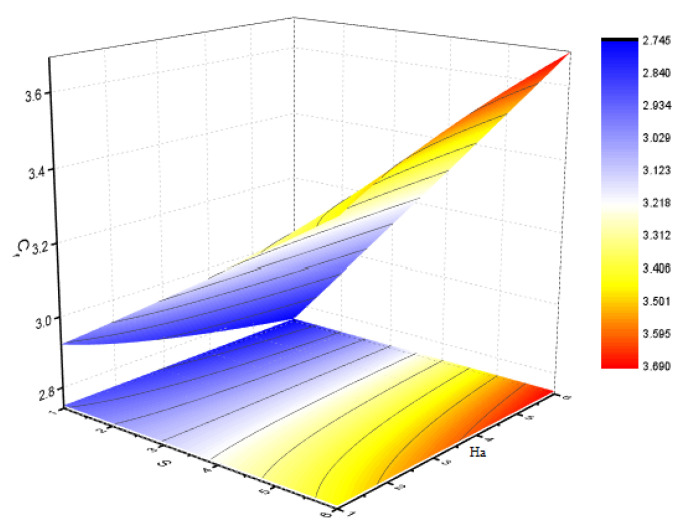
Effect of skin friction for *S* and Ha and fixed values of $Re_{m}=2.5,Pr=6.2,Ec=0.5,\delta =0.4,\Phi_{1}=0.02,m=0.3$.

**Figure 10 nanomaterials-12-00180-f010:**
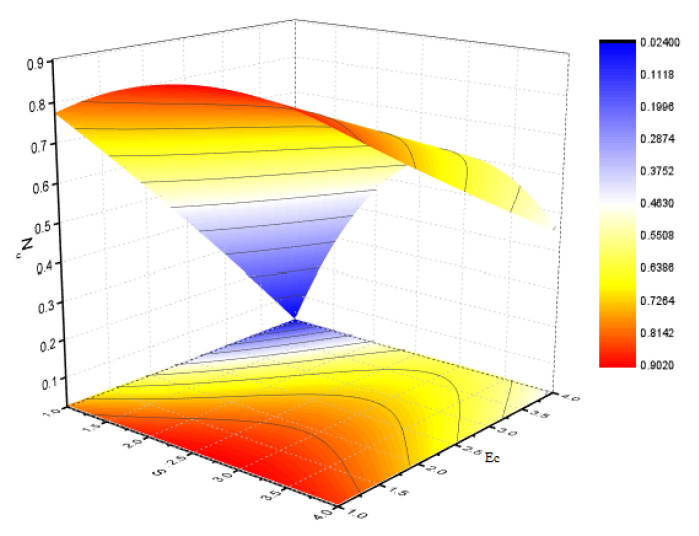
Effect of Nusslet number for *S* and Ec and fixed values of $Re_{m}=2.5,Pr=6.2,\delta =0.4,\Phi_{1}=0.02,m=0.3$.

**Table 1 nanomaterials-12-00180-t001:** The thermophysical properties of water base fluid and hybrid nanoparticles.

	ρ	$C_{p}$	κ	σ
$H_{2}O$	997.1	4179	0.613	5.5 × $10^{-6}$
$Fe_{3}O_{4}$	5200	670	6	9.74 × $10^{6}$
SWCNT	2600	425	6600	$10^{6}$
MWCNT	1600	796	3000	$10^{7}$

**Table 2 nanomaterials-12-00180-t002:** Comparison of the numerical results of $f^{\prime \prime }\left(1\right)$ when $\Phi_{1}$ = $\Phi_{2}$ = $Re_{m}$ = δ = 0.

*M*	*S*	M. Bilal et al. [15]	Present
0	0.5	4.713254	4.713061
1		4.739148	4.739224
2		4.820361	4.820101
3		4.396271	4.396400
2	0	1.842331	1.842350
	0.3	3.653601	3.653573
	0.6	5.391148	5.391053
	1	7.593006	7.593187

**Table 3 nanomaterials-12-00180-t003:** Comparison of the numerical results by two methods PCM and BVP4C for Skin friction and Nusselt number, with various physical parameters and $\Phi_{1}$ = $\Phi_{2}$ = 0.1.

	PCM	BVP4C	PCM	BVP4C
S	$f^{\prime \prime }\left(1\right)$	$f^{\prime \prime }\left(1\right)$	$-\theta^{\prime }\left(1\right)$	$-\theta^{\prime }\left(1\right)$
0.1	5.9662	5.9917	2.7615	2.7688
0.2	5.9105	5.9143	2.7369	2.7344
0.3	5.8437	5.8422	2.7103	2.7178
0.4	5.7714	5.7756	2.6831	2.6812
0.5	5.6968	5.6990	2.6557	2.6554
0.6	5.6219	5.6252	2.6278	2.6282
0.7	5.5489	5.5432	2.6057	2.6076
0.8	5.4779	5.4723	2.5822	2.5871
0.9	5.4081	5.4023	2.5538	2.5532

**Table 4 nanomaterials-12-00180-t004:** Comparison of the numerical results by two methods PCM and BVP4C for skin friction and Nusselt number, with various physical parameters and $\Phi_{2}$ = 0.02.

	PCM	BVP4C	PCM	BVP4C
$\Phi_{1}$	$f^{\prime \prime }\left(1\right)$	$f^{\prime \prime }\left(1\right)$	$-\theta^{\prime }\left(1\right)$	$-\theta^{\prime }\left(1\right)$
0.1	5.1032	5.1076	0.6308	0.6366
0.2	5.0736	5.0722	0.6774	0.6732
0.3	5.0801	5.0854	0.7399	0.7321
0.4	5.1222	5.1200	0.7957	0.7960
0.5	5.1989	5.1943	0.7765	0.7790
0.6	5.3095	5.3083	0.4537	0.4502
0.7	5.4558	5.4511	0.3872	0.3845
0.8	5.6343	5.6334	0.2733	0.2712
0.9	5.8421	5.8466	0.1209	0.1255

**Table 5 nanomaterials-12-00180-t005:** Comparison of the numerical results by two methods PCM and BVP4C for skin friction and Nusselt number, with various physical parameters and $\Phi_{1}$ = $\Phi_{2}$ = 0.1.

	PCM	BVP4C	PCM	BVP4C
Ha	$f^{\prime \prime }\left(1\right)$	$f^{\prime \prime }\left(1\right)$	$-\theta^{\prime }\left(1\right)$	$-\theta^{\prime }\left(1\right)$
0.1	5.9632	5.9632	2.7584	2.7584
0.2	5.9391	5.9391	2.7486	2.7486
0.3	5.8992	5.8992	2.7325	2.7325
0.4	5.8437	5.8437	2.7103	2.7103
0.5	5.7731	5.7731	2.6826	2.6826
0.6	5.6879	5.6879	2.6499	2.6499
0.7	5.5887	5.5887	2.6128	2.6128
0.8	5.4762	5.4762	2.5721	2.5721
0.9	5.3512	5.3512	2.5283	2.5283

**Table 6 nanomaterials-12-00180-t006:** Numerical results by parametric continuation method for hybrid nanofluid with various physical parameters.

	$H_{2}O$, ${\mathrm{Fe}}_{3}O_{4}$,	SWCNT	$H_{2}O$, ${\mathrm{Fe}}_{3}O_{4}$,	MWCNT	$H_{2}O,{\mathrm{Fe}}_{3}O_{4}$	
$\Phi_{2}$	$f^{\prime \prime }\left(1\right)$	$-\theta^{\prime }\left(1\right)$	$f^{\prime \prime }\left(1\right)$	$-\theta^{\prime }\left(1\right)$	$f^{\prime \prime }\left(1\right)$	$-\theta^{\prime }\left(1\right)$
0.02	5.6736	0.1161	5.6834	0.1351	5.6997	0.0573
0.03	5.6725	0.1170	5.6822	0.1356	5.6984	0.0564
0.04	5.6717	0.1181	5.6811	0.1361	5.6965	0.0554
0.05	5.6704	0.1196	5.6802	0.1370	5.6953	0.0542
0.06	5.6691	0.1208	5.6793	0.1379	5.6940	0.0529
0.07	5.6679	0.1220	5.6780	0.1387	5.6926	0.0516
0.08	5.6662	0.1236	5.6766	0.1399	5.6911	0.0503
0.09	5.6645	0.1251	5.6752	0.1410	5.6891	0.0488
0.10	5.6633	0.1269	5.6732	0.1422	5.6873	0.0471
0.11	5.6619	0.1282	5.6715	0.1438	5.6858	0.0457
0.12	5.6603	0.1301	5.6690	0.1455	5.6843	0.0441
0.13	5.6593	0.1325	5.6672	0.1471	5.6829	0.0425

**Table 7 nanomaterials-12-00180-t007:** Numerical results by BVP4C Method for hybrid nanofluid with various physical parameters.

	$H_{2}O$, ${\mathrm{Fe}}_{3}O_{4}$,	SWCNT	$H_{2}O$, ${\mathrm{Fe}}_{3}O_{4}$,	MWCNT	$H_{2}O,{\mathrm{Fe}}_{3}O_{4}$	
$\Phi_{2}$	$f^{\prime \prime }\left(1\right)$	$-\theta^{\prime }\left(1\right)$	$f^{\prime \prime }\left(1\right)$	$-\theta^{\prime }\left(1\right)$	$f^{\prime \prime }\left(1\right)$	$-\theta^{\prime }\left(1\right)$
0.02	5.6731	0.1157	5.6843	0.1343	5.6980	0.0563
0.03	5.6721	0.1164	5.6830	0.1350	5.6971	0.0551
0.04	5.6711	0.1173	5.6821	0.1355	5.6961	0.0540
0.05	5.6698	0.1190	5.6810	0.1363	5.6945	0.0533
0.06	5.6683	0.1201	5.6800	0.1370	5.6930	0.0521
0.07	5.6671	0.1213	5.6791	0.1378	5.6918	0.0502
0.08	5.6653	0.1226	5.6773	0.1391	5.6902	0.0492
0.09	5.6636	0.1242	5.6764	0.1403	5.6887	0.0476
0.10	5.6622	0.1253	5.6750	0.1413	5.6871	0.0462
0.11	5.6610	0.1270	5.6730	0.1428	5.6847	0.0445
0.12	5.6595	0.1292	5.6708	0.1441	5.6828	0.0432
0.13	5.6580	0.1311	5.6691	0.1464	5.6814	0.0413

**Table 8 nanomaterials-12-00180-t008:** The heat transfer has been calculated percent wise as for the various nanoparticles *Pr* = 6.2, *S* = 1.5, *Ec* = 0.4, using the percentage formula %increase = $\frac{With\;Nanoparticle}{Without\;Nanoparticle}$× 100 = Result, Result-100 = %enhancement.

$\Phi_{1},\;\Phi_{2}$	$-\theta^{\prime }\left(1\right)\;\mathrm{for}\;{\mathrm{Fe}}_{3}O_{4},\mathrm{SWCNT}$	$-\theta^{\prime }\left(1\right)\;\mathrm{for}\;{\mathrm{Fe}}_{3}O_{4},\mathrm{MWCNT}$
0.0	5.5582	5.5582
0.01	5.6490	5.6561
	**(1.6336% increase)**	**(1.7614% increase)**
0.03	5.8421	5.8628
	**(5.1078% increase)**	**(5.4802% increase)**
0.04	5.9446	5.9718
	**(6.9519% increase)**	**(7.4413% increase)**

## Data Availability

Not Applicable.

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
