# Peer review of "Numerical Analysis of Unsteady Hybrid Nanofluid Flow Comprising CNTs-Ferrousoxide/Water with Variable Magnetic Field"

_nanomaterials, 2022, doi:10.3390/nano12020180_

Round 1

Reviewer 1 Report

The work well done but have some  corrections needed to improve the quality of the presentation. I enlist them below:

  • Why similarity variable contain only one variable? Explanation and justification is needed.
  • The Eckert number Ec is a number (constant). However, in this paper Ec is a function of the indpendent variable x, Therefore, the partial differential equations cannot bereduced to the ordinary (similarity) equations . Justification is to be given.
  • Highlight the article innovation point in the abstract. Why should anyone read your article?
  • please present shortly novelty of your work at the first sentence of abstract section.  please add references to all of equations.
  • please valid your method with another numerical approaches.
  • To enrich the introduction section, the following published papers can be cited:

Assessment of the local thermal non-equilibrium condition for nanofluid flow through porous media: a comparative analysis, Physical impact of thermo-diffusion and diffusion-thermo on Marangoni convective flow of hybrid nanofluid (MnZiFe2O4–NiZnFe2O4–H2O) with nonlinear heat …, Two-Phase Darcy-Forchheimer Flow of Dusty Hybrid Nanofluid with Viscous Dissipation Over a Cylinder, Comparative analysis of (Zinc ferrite, Nickel Zinc ferrite) hybrid nanofluids slip flow with entropy generation, Non-Newtonian hybrid nanofluid flow over vertically upward/downward moving rotating disk in a Darcy–Forchheimer porous medium,

  • The conclusion section is not proper and has to be totally revised. This section should clearly demonstrate the results by percentage and compare them by data and numbers to show the best scenario
  • English must be improved. There are typos and grammatical errors within the text that has not been fixed yet.
  • Some references are older than 2015 and therefore, they are abolished. May the reviewer ask the authors to change these references to newer ones? Some suggested papers are as below;

Marangoni convective flow of hybrid nanofluid (MnZnFe2O4-NiZnFe2O4-H2O) with Darcy Forchheimer medium,

Two-phase flow of dusty fluid with suspended hybrid nanoparticles over a stretching cylinder with modified Fourier heat flux,

  • Motivation of the study is missing.
  • Punctuations are used randomly. Insert comma or full stop after each and every equation accordingly

Author Response

REPLY TO REVIEWER 1

Comments and Suggestions for Authors

The work well done but have some corrections needed to improve the quality of the presentation. I enlist them below

Q1: Why similarity variable contains only one variable? Explanation and justification are needed.

Ans: The similarity variables contain only one variable y because from the problem geometry on page # 5, the fluid motion is only due to the motion of the upper plate which is along y-axis. It means that all the variables will be y dependent. Also, the squeezing phenomenon is time dependent phenomenon, so, this problem is also time dependent. All these variables and their dependence are clearly written on page # 7 in equation # 9.

Q2: The Eckert number Ec is a number (constant). However, in this paper Ec is a function of the independent variable x, Therefore, the partial differential equations cannot be reduced to the ordinary (similarity) equations. Justification is to be given.

Ans: The expression for Eckert number was written mistakenly. Correction has been made. (See with color red on page 8).

Q3:  Highlight the article innovation point in the abstract. Why should anyone read your article?

Ans: The innovation point and reasons for the study of this article has been added in the abstract. (See with color red on page 1).

Q4: please present shortly novelty of your work at the first sentence of abstract section.  please add references to all of equations.

Ans: Novelty of the problem has been added in the abstract. Also, some explanation of the novelty has been added at the end of the introduction section. (See with color red on page 1 and 5).

Q5: please valid your method with another numerical approaches.

Ans: The validity of the method has been proved by tables 3 to 7. Also, some new tables are added to strengthen this article. Table (3-7) indicates validity of the solution of PCM. Suggestion has been honored.  (See with color red on page 9-13).

Q6: To enrich the introduction section, the following published papers can be cited:

Assessment of the local thermal non-equilibrium condition for nanofluid flow through porous media: a comparative analysis, Physical impact of thermo-diffusion and diffusion-thermo on Marangoni convective flow of hybrid nanofluid (MnZiFe2O4–NiZnFe2O4–H2O) with nonlinear heat …, Two-Phase Darcy-Forchheimer Flow of Dusty Hybrid Nanofluid with Viscous Dissipation Over a Cylinder, Comparative analysis of (Zinc ferrite, Nickel Zinc ferrite) hybrid nanofluids slip flow with entropy generation, Non-Newtonian hybrid nanofluid flow over vertically upward/downward moving rotating disk in a Darcy–Forchheimer porous medium,

Ans: All the above-mentioned articles have been cited in the introduction section. (See with color red on page 4 and 5).

Q7: The conclusion section is not proper and has to be totally revised. This section should clearly demonstrate the results by percentage and compare them by data and numbers to show the best scenario

Ans: Conclusion section has been totally revised. All the derived results are now written in a very clear manner. (See with color red on page 16).

Q8: English must be improved. There are typos and grammatical errors within the text that has not been fixed yet.

Ans: The English language has been improved by a native speaker. All the typo and grammatical mistakes has been corrected according to the best of our knowledge.

Q9: Some references are older than 2015 and therefore, they are abolished. May the reviewer ask the authors to change these references to newer ones? Some suggested papers are as below;

Marangoni convective flow of hybrid nanofluid (MnZnFe2O4-NiZnFe2O4-H2O) with Darcy Forchheimer medium,

Two-phase flow of dusty fluid with suspended hybrid nanoparticles over a stretching cylinder with modified Fourier heat flux,

Ans: These references have been added in the introduction section. (See with color red on page 4 and 5).

Q10: Motivation of the study is missing.

Ans: Motivation of the study has been added both in the abstract and introduction section. (See with color red on page 1 and 2).

Q11: Punctuations are used randomly. Insert comma or full stop after each equation accordingly

Ans: All the typos and grammatical mistakes have been corrected by a native speaker. Punctuations are also corrected.

Reviewer 2 Report

-In abstract, before layout of your purpose, try to add a sentence at the beginning with a brief background information description. There are too many simple conclusive statements in the abstract, which are well-known to people who are involved in this area. You should first layout a background information description.

-Second, point out what is the most important in the current field of research and what has not yet been solved. Third, explain your novel method in detail how you solve this problem. Last, a brief description of your own findings should be presented. If your methods and findings are novel and interesting to people who are involved in this area, they will continue to read your main text, or they will not waste their time.

-It is amazing that very limited novel and interesting information can be found after reading such a long abstract. Besides, they are similar to the conclusion part. Your paper gives readers the impression that the research is too superficial and not in-depth.

-Key words: these key words are too general. Where is your novel point? Highlighting your novel study methods and attractive findings in key words is necessary. Your key words can also be adopted by any paper sharing the same topic. So, which key word can distinguish your paper from theirs?

-In Introduction, the authors presented too much conclusive sentences without any data or references to validate them. I can find a large number of papers to argue them. If a sentence in which a conclusion is made, a reference should be added to it unless the conclusion itself is yours. Did you write them casually without referring to previous papers? If not, please add references to it. Similar problems should be noticed throughout the introduction. Reading your paper makes people feel uncomfortable because your paper is written in a casual and not unrigorous way.

  • Introduction should be updated by adding new papers such as: 

    https://doi.org/10.1016/j.ijheatmasstransfer.2021.122232,

    https://doi.org/10.1016/j.icheatmasstransfer.2021.105648,

    https://doi.org/10.1016/j.physrep.2021.07.002

  •  

Author Response

REPLY TO REVIEWER 2

Comments and Suggestions for Authors

Q1. In abstract, before layout of your purpose, try to add a sentence at the beginning with a brief background information description. There are too many simple conclusive statements in the abstract, which are well-known to people who are involved in this area. You should first layout a background information description.

Ans. Introduction of hybrid nano-fluid, applications of nano-fluid, a brief description and novelty of the problem have been added in the abstract. Also, the main results are added in clear words. The abstract has been totally revised and the suggestion has been honored. (See with colored red on page 1)

Q2. Second, point out what is the most important in the current field of research and what has not yet been solved. Third, explain your novel method in detail how you solve this problem. Last, a brief description of your own findings should be presented. If your methods and findings are novel and interesting to people who are involved in this area, they will continue to read your main text, or they will not waste their time.

Ans. Importance of the problem and the current research needs have been added in the abstract and at the start of the introduction section. The method has been explained step by step in detail on page # 9. Also, the code of this method has been shared for the reader’s help on  page 23. The main findings are revised in the conclusion section and some of them are also added in the abstract.

Q3. It is amazing that very limited novel and interesting information can be found after reading such a long abstract. Besides, they are similar to the conclusion part. Your paper gives readers the impression that the research is too superficial and not in-depth.

Ans. Novelty of the problem has been added at the last paragraph of the introduction section. The abstract has been revised for the better understanding of the article. (See with colored red on page 1 and 5).

Q4. Key words: these key words are too general. Where is your novel point? Highlighting your novel study methods and attractive findings in key words is necessary. Your key words can also be adopted by any paper sharing the same topic. So, which key word can distinguish your paper from theirs?

Ans. All the key words have been revised. The magnetic Reynold number, variable magnetic field and the PCM are the newly introduced key words and are also new in the literature. Hope these key words will also attract readers to read the article.

Q5. In Introduction, the authors presented too much conclusive sentences without any data or references to validate them. I can find a large number of papers to argue them. If a sentence in which a conclusion is made, a reference should be added to it unless the conclusion itself is yours. Did you write them casually without referring to previous papers? If not, please add references to it. Similar problems should be noticed throughout the introduction. Reading your paper makes people feel uncomfortable because your paper is written in a casual and not unrigorous way.

Ans. All the casually written papers has been deleted. The introduction has been also revised. All the conclusive sentences are corrected. (See with colored red on pages 2- 4).

Q6. Introduction should be updated by adding new papers such as: 

https://doi.org/10.1016/j.ijheatmasstransfer.2021.122232,

https://doi.org/10.1016/j.icheatmasstransfer.2021.105648,

https://doi.org/10.1016/j.physrep.2021.07.002

Ans. Introduction section has been updated by adding all these new papers. (See with colored red on page 21-22-23).

Reviewer 3 Report

Title: 

Numerical Analysis of Unsteady Hybrid Nanouid Flow comprising CNTs-Ferrousoxide/Water with Variable Magnetic Field

COMMENTS FOR THE AUTHOR:

Reviewer #: This is a potentially interesting paper that addresses an important TRANSPORT  problem. However, before I can recommend this paper for publication, it should be revised subject to the following suggestions.

  1. Pls include  units in the nomenclature section.
  2. Include references for eqn. 9.
  3. Include derivation of magnetic filed equations  in the appendix for the readers
  4. Illustrate numerical methods and include code in the appendix for readers.
  5. Include future works
  6. Physical reason behind the trend os the graph should be included.
  7. Many parameters such as  Ec, delta are space and time dependent which violates the similarity.
  8. Graphs of physical quantites should be included.
  9. Reduce old references and include following recent refs.

Numerical solution of bio-nano-convection transport from a horizontal plate with blowing and multiple slip effects

Magnetohydrodynamic bio-nano-convective slip flow with Stefan blowing effects over a rotating disc

Comments: major revision and re-review

Author Response

REPLY TO REVIEWER 3

COMMENTS FOR THE AUTHOR:

Reviewer #: This is a potentially interesting paper that addresses an important TRANSPORT  problem. However, before I can recommend this paper for publication, it should be revised subject to the following suggestions.

Q1. Pls include units in the nomenclature section.

Ans. A nomenclature table has been added. (See with colored red on 21)

Q2. Include references for eqn. 9.

Ans. References of equation 9 has been added. (See with colored red on page 6)

Q3. Include derivation of magnetic filed equations  in the appendix for the readers

Ans. Dear reviewer the derivation of the magnetic field equations is a too long process. Theses derivations took ten pages to complete. Adding these pages to the manuscript will make it too long as there are already 25 pages. If you don’t mind, we can email you the proof of these equations. Will happily wait for your positive response. Thank You

Q4. Illustrate numerical methods and include code in the appendix for readers.

Ans. The code of the PCM has been added in the appendix. (See with colored red on pages 21-25)

Q5. Include future works

Ans. The future work has been added in the conclusion section. (See with colored red on page 19)

Q6. Physical reason behind the trend of the graph should be included.

Ans. Most of the Results and Discussion section have been revised by adding the physical reasons of the trends. (See with colored red on pages 14-15 )

Q7. Many parameters such as Ec, delta are space and time dependent which violates the similarity.

Ans. Eckert number was written mistakenly. The mistake has been corrected.  (See with colored red on page 8)

Q8. Graphs of physical quantites should be included.

Ans. Graphs of the physical quantities have been added. (See with colored red on page  )

Q9. Reduce old references and include following recent refs.

Numerical solution of bio-nano-convection transport from a horizontal plate with blowing and multiple slip effects

Magnetohydrodynamic bio-nano-convective slip flow with Stefan blowing effects over a rotating disc

Ans. All the old references have been deleted and the suggested papers are included in the introduction section. (See with colored red on page 22,23)

Round 2

Reviewer 1 Report

The authors have addressed the queries at the satisfactory level and revised the article as suggested. Hence, I recommend the publication 

Reviewer 3 Report

Title: 

Numerical Analysis of Unsteady Hybrid Nanouid Flow comprising CNTs-Ferrousoxide/Water with Variable Magnetic Field

COMMENTS FOR THE AUTHOR:

Reviewer #: This is a potentially interesting paper that addresses an important TRANSPORT  problem. However, before I can recommend this paper for publication, it should be revised subject to the following suggestions.

  1. Pls include  units in the nomenclature section.
  2. Include references for eqn. 9.
  3. Include derivation of magnetic filed equations  in the appendix for the readers
  4. Illustrate numerical methods and include code in the appendix for readers.
  5. Include future works
  6. Physical reason behind the trend os the graph should be included.
  7. Many parameters such as  Ec, delta are space and time dependent which violates the similarity.
  8. Graphs of physical quantites should be included.
  9. Reduce old references and include following recent refs.

Numerical solution of bio-nano-convection transport from a horizontal plate with blowing and multiple slip effects

Magnetohydrodynamic bio-nano-convective slip flow with Stefan blowing effects over a rotating disc